# Diabetes Mellitus and Heart Failure

**DOI:** 10.3390/jpm12101698

**Published:** 2022-10-11

**Authors:** Wun-Zhih Siao, Yong-Hsin Chen, Chin-Feng Tsai, Chun-Ming Lee, Gwo-Ping Jong

**Affiliations:** 1Division of Cardiology, Department of Internal Medicine, Chung Shan Medical University Hospital, Taichung 40201, Taiwan; 2Institute of Medicine, College of Medicine, Chung Shan Medical University, Taichung 40201, Taiwan; 3Department of Public Health, Chung Shan Medical University, Taichung 40201, Taiwan; 4Department of Occupational Safety and Health, Chung Shan Medical University Hospital, Taichung 40201, Taiwan; 5Department of Internal Medicine, St. Joseph’s Hospital, Yunlin County 63201, Taiwan

**Keywords:** diabetes mellitus, heart failure, sodium–glucose cotransporter 2 inhibitors, reduced ejection fraction, preserved ejection fraction

## Abstract

The coexistence of diabetes mellitus (DM) and heart failure (HF) is frequent and is associated with a higher risk of hospitalization for HF and all-cause and cardiovascular mortality. It has been estimated that millions of people are affected by HF and DM, and the prevalence of both conditions has increased over time. Concomitant HF and diabetes confer a worse prognosis than each alone; therefore, managing DM care is critical for preventing HF. This article reviews the prevalence of HF and diabetes and the correlated prognosis as well as provides a basic understanding of diabetic cardiomyopathy, including its pathophysiology, focusing on the relationship between DM and HF with a preserved ejection fraction and summarizes the potential aldosterone and the mineralocorticoid receptor antagonists approaches for managing heart failure and DM. Sodium–glucose cotransporter 2 inhibitors (SGLT2Is) are an emerging class of glucose-lowering drugs, and the role of SGLT2Is in DM patients with HF was reviewed to establish updated and comprehensive concepts for improving optimal medical care in clinical practice.

## 1. Introduction

Currently, heart failure (HF) affects at least 26 million people worldwide and is increasing in prevalence [1]. Similarly, the prevalence of diabetes mellitus (DM) continues to increase over time [1]. Concomitant HF and DM generate enormous clinical and economic burdens. Evidence from observational studies confirms HF and DM as independent risk factors of each other [2,3]. Concomitant HF and DM also lead to a worse prognosis than HF or DM alone. Thus, it is important to identify and manage concomitant HF and DM effectively.

This article reviews the prevalence of HF and DM and their correlation with prognosis. This review also provides emergency and intensive care service providers with a basic understanding of diabetic cardiomyopathy, including its pathophysiology, with a focus on the relationship of DM and HF with preserved ejection fraction (HFpEF). The role of sodium–glucose cotransporter 2 inhibitors in diabetic patients with HF is also reviewed to establish updated and comprehensive concepts for the improvement of optimal medical care in clinical practice.

## 2. Epidemiology

The prevalence of DM among patients with HF with reduced ejection fraction (HFrEF) ranged from 10 to 30% [4]. HF, especially in the advanced stage, has also been regarded as a predictor of DM development among older people [5]. The Framingham study [6] demonstrated a 2.4- to 5-fold increase in the incidence of HF in patients with DM. Particularly affected are older people suffering from DM in the long term, with their insulin use or low body mass index predicting HF occurrence. Specifically, poor glycemic control correlated with an increased risk of HF, with a 1% elevation in hemoglobin A1c (HbA1c) equating to an 8% increment in HF risk [7]. Patients with HF and poor glycemic control (HbA1c > 8%) were observed to suffer from poor cardiovascular outcomes in their lifetime [8]. Meanwhile, about half of patients admitted with HF have impaired glucose tolerance or are newly diagnosed with DM. Relative to normoglycemia, impaired glucose tolerance and DM are associated with all-cause and cardiovascular mortality [9].

A prospective cohort study described the relatively low prevalence (3.7%) and incidence (0.02% per year) of HF and predominant diastolic HF in patients with type 1 DM, specifically those with concurrent hypertension or coronary artery disease. The relatively young study population and common use of intensive insulin therapy and concomitant statin and aspirin treatment might have contributed to the difference between the study’s prevalence and incidence and those reported for patients with type 2 DM [10]. In addition, patients with type 1 DM and albuminuria are at a higher risk of systolic dysfunction than those with normoalbuminuria [11].

With regard to prognosis, patients with concomitant DM and HF have higher mortality rates than those without DM or HF [12,13]. A prospective cohort study showed a 33% increased risk of hospitalization for HF (HHF) among patients with DM relative to their counterparts without DM [14]. In a biomarker substudy of PARADIGM-HF [15], troponin T (TnT) and N-terminal (NT)-pro hormone BNP (NT-proBNP) were reported as independent predictors of adverse outcomes for HF patients with DM. The study found higher concentrations of TnT in HFrEF patients with DM than in those without DM regardless of whether HF was of ischemic or non-ischemic etiology. In addition, the elevation of TnT (≥18 ng/L) and NT-proBNP levels suggested an increased risk of cardiovascular death or HHF.

## 3. Heart Failure with Preserved Ejection Fraction and Diabetes

Although HF with systolic dysfunction is commonly mentioned in clinical trials, HFpEF has raised increasing attention recently. Observation studies have demonstrated that diastolic dysfunction is prevalent in types 1 and 2 DM [16,17]. Left ventricular (LV) diastolic dysfunction is also common in patients with pre-DM [18]. In addition, LV diastolic dysfunction has been observed in >50% of patients with asymptomatic DM, and its severity corresponds to insulin resistance [19]. Meanwhile, DM is a common comorbidity and has a 45% prevalence in patients with HFpEF, especially those with new-onset HFpEF [20]. Existing research has investigated the relationship between DM and its comorbidity, and the results indicate that relative to patients without DM, those with HFpEF and DM have a high body mass index, young age, hypertension, ischemic heart disease, and symptoms of fluid overload [21,22,23,24]. Data from the Get With The Guidelines-Heart Failure registry indicate that patients with HFpEF and DM tend to be young and male and are likely to suffer from various comorbidities, including hyperlipidemia, chronic obstructive pulmonary disease, ischemic cerebrovascular accident, peripheral vascular disease, renal insufficiency, anemia, and depression [25].

With regard to prognosis, DM exerts a negative prognostic effect on patients with HFpEF. Specifically, patients with HFpEF and DM face a higher risk for HHF and cardiovascular death than those suffering from HFpEF without DM [21,22,26]. Moreover, cluster analysis showed that DM with obesity, hypertension, and diastolic dysfunction, or DM with LV hypertrophy and systolic dysfunction, have a worse prognosis than isolated DM with preserved systolic and diastolic function [23]. Insulin-treated DM also predicts sudden death in HFpEF [27]. In the Candesartan in Heart Failure: Assessment of Reduction in Mortality and Morbidity trial, patients with DM and HFpEF showed a lower mortality rate than those with HFrEF [21]. This difference may be attributed to the relatively high prevalence of ischemic heart disease in patients with HFrEF.

The symptoms of diastolic dysfunction in patients with DM are often nonspecific and thus hamper early detection. Holland et al. described echocardiography as a useful tool to uncover potential diastolic dysfunction in DM. Increased E/eʹ during stress provides early information and prognosis value in the early stage of diastolic dysfunction [28]. Over time, E/eʹ at rest and even the elevation of the left ventricle filling pressure occurs.

DM-associated HF is a complex entity involving multiple factors. DM is a documented and strong risk factor for coronary artery disease with consequent ischemic heart disease. Ischemic events are common in the diabetic population, and they mainly contribute to HF development. Meanwhile, diabetic cardiomyopathy is a unique HF subtype in the absence of cardiac ischemia or other well-established risk factors, such as hypertension. The structural and functional dysfunctions of diabetic cardiomyopathy primarily result from the insulin resistance of diabetic myocardium and consequent hyperinsulinemia. Moreover, hyperinsulinemia can induce inappropriate activation of the systemic and cardiac tissue renin–angiotensin–aldosterone system; although a state of salt and volume excess contributes to the development of diabetic cardiomyopathy. Although no clear definition describes diabetic cardiomyopathy, it currently has two different subtypes: hypertrophic dominant with preserved ejection fraction and dilative dominant with reduced ejection fraction [29].

Myocardial hypertrophy is a common finding in diabetic hearts and a result of oxidative stress, inflammation, and upregulation of the renin–angiotensin and endothelia systems, leading to cardiac stiffness. The extent of myocardial hypertrophy correlated with glycemic control [30]. Systolic dysfunction tends to occur in the later stage of DM, whereas diastolic dysfunction occurs in the early stages of the disease course [19,31]. However, the echocardiographic characteristics of systolic strain alteration have been observed in 28% of patients with DM and normal diastolic function [32].

Glucose metabolism plays a crucial role in the development of myocardial dysfunction. Advanced glycation end products (AGEs) are glycated after exposure to hyperglycaemia, and are produced from non-enzymatic glycosylation of lipids, lipoproteins and amino acids. AGEs cause increased myocardial stiffness through the formation of crosslinks in collagens and laminins, subsequently increasing fibrosis and thereby reducing cardiac compliance and LV diastolic dysfunction [33,34]. In addition, AGEs may bind to the cell surface receptor for AGE to prompt the increased expression of TGF-β, connective tissue growth factor, or poly (ADP-ribose) polymerase 1; and the reduced expression of MMP-2 contribute to the dysregulation of extracellular matrix degradation and result in increased fibrosis in diabetic myocardium [35,36].

The other glucose metabolism product, β-N-acetylglucosamine (O-GlcNAc), results in diabetic myocardium dysfunction through the modification of the Ca2þ/calmodulin-dependent protein kinase II, phospholamban, and myofilaments; these factors negatively affect cardiac systolic and diastolic function. In addition, O-GlcNAc can bind to mitochondrial proteins, impair mitochondrial function, and subsequently increase the production of ROS, which promotes inflammation and fibrosis [35,37].

Apart from glucose metabolism, several mechanisms have been identified to contribute to myocardial fibrosis, cardiomyocyte injury, and remodeling in DM, such as the activation of the renin–angiotensin–aldosterone system and sympathetic activity, altered myocardial insulin signaling, mitochondrial dysfunction, gene dysregulation of microRNAs and transcription factors, epigenetic modifications, oxidative stress, and endoplasmic reticulum stress [32]. Lipotoxicity, which is described as the increased circulating levels of fatty acid deposition in diabetic hearts, leads to increased myocardial oxygen consumption, reduced cardiac efficiency, and myocardial cell damage [38]. Impaired mitochondrial function in the setting of DM also contributes to the generation of reactive oxygen species, which in turn enhances the progression of myocardial dysfunction [39]. Increased myocardial fibrosis and inflammation status contribute to the activation of cellular proapoptotic signaling pathways with resultant myocardial cell death [32].

## 4. Mineralocorticoid Receptor Antagonists in DM and HF

There has been a dramatic increase in knowledge about the role of aldosterone and mineralocorticoid receptor (MR) antagonists (MRAs) in the pathophysiology of cardiovascular diseases in recent years [40]. The MR is the principal mediator of the effect of aldosterone on renal sodium reabsorption in the distal nephron and systemic modulators of extracellular matrix, inflammation, and fibrosis. MRAs are included in treatment paradigms for HFrEF under the RALES trial (for spironolactone) [41] and the EMPHASIS-HF trial (for eplerenone) [42]. Furthermore, the 2019 ESC guidelines for the treatment of acute and chronic HF continue to give a class I and level of recommendation A to MRAs in the treatment of chronic HFrEF [43]. In the recent 3 years, the nonsteroidal MRA finerenone was also shown to effectively reduce the risk of HHF in patients with type 2 DM and chronic kidney disease in two clinical trials (FIDELIO-DKD and FIGARO-DKD) [44,45].

## 5. Sodium–Glucose Cotransporter 2 Inhibitors in Diabetes and Heart Failure

Four SGLT2is (i.e., empagliflozin, dapagliflozin, canagliflozin, and ertugliflozin) are approved by the European Medicines Agency and the US Food and Drug Administration on the basis of large-scale randomized controlled trials (RCTs) and documented cardiovascular benefits. The cardiovascular benefits of SGLT2is are beyond glycemic control. Several mechanisms contribute to the cardioprotective effect of SGLT2is, and they include the reduction in preload and afterload secondary to natriuresis and a decrease in blood pressure, ketone body-based myocardial metabolism, and improvement of myocardial remodeling [46]. Selected mechanisms of HF and renoprotective benefits of SGLT2is are schematized in Figure 1. The antiarrhythmic effect among diabetic patients treated with SGLT2is was also revealed by a population-based cohort study [47].

Substantial RCTs have also established the cardioprotective role of SGLT2is. In particular, the EMPA-REG OUTCOME (empagliflozin), CANVAS (canagliflozin), DECLARE-TIMI 58 (dapagliflozin), and VERTIS-CV (ertugliflozin) trials showed a significant reduction in HFH among patients with diabetes treated with SGLT2is [48,49,50,51]. The enrolled populations in EMPA-REG OUTCOME and VERTIS-CV were those with established cardiovascular disease; those in CANVAS and DECLARE-TIMI 58 were at high risk for or suffering from atherosclerotic cardiovascular disease. In the EMPA-REG OUTCOME trial, the use of empagliflozin reduced the risk of cardiovascular death. Meanwhile, the result of the diabetes treatment with SGLT2is in the CANVAS, DECLARE-TIMI 58, and VERTIS-CV trials did not reach statistical significance in relation to cardiovascular death. Randomized clinical trials investigating cardiovascular outcomes with SGLT2i therapies are summarized in Table 1.

Data from clinical practice are consistent with large-scale RCTs. The CVD-Real [52], OBSERVE-4D [53], and EMPRISE [54] studies found that patients with type 2 diabetes treated with SGLT2is have a lower risk of HHF than their counterparts treated with other glucose-lowering agents.

Additionally, the Canagliflozin and Renal Events in Diabetes and Established Nephropathy Clinical Evaluation (CREDENCE) trial (data from the designated kidney outcome trial with canagliflozin in patients with T2D) and the dapagliflozin in patients with chronic kidney disease (DAPA-CKD) trial (data from the designated kidney outcome trial with dapagliflozin in patients with or without T2D) also showed an impressive reduction in the risk of HHF.

## 6. Sodium–Glucose Cotransporter 2 Inhibitors in Heart Failure with Reduced Ejection Fraction

Several studies have investigated the impact of SGLT2is in diabetic patients with HF stratified by reduced or preserved ejection fraction. The DAPA-HF trial [55] demonstrated a significant reduction in HFH among patients with reduced ejection fraction treated with dapagliflozin relative to those under placebo treatment. An improvement in quality of life, as measured using KCCQ, was observed in the dapagliflozin-treated group. The risk reduction in the first and recurrent HFH was also observed in the EMPEROR-Reduced trial [56], which recruited subjects treated with empagliflozin with a reduced ejection fraction of 40% or less and with New York Heart Association class II–IV symptoms. The benefits of HHF reduction were consistent in the DAPA-HF trial or EMPEROR-Reduced trial regardless of the presence of diabetes. SGLT2I treatment in HF patients was added to the optimized guideline-directed medical therapy in the DAPA-HF trial or EMPEROR-Reduced trial.

A prespecified meta-analysis of the DAPA-HF and EMPEROR-Reduced trials [57] aimed to investigate the effects of SGLT2i treatment on cardiovascular outcomes in patients with HF and reduced ejection. The meta-analysis showed a 14% reduction in cardiovascular death (pooled HR: 0.86, CI: 0.76–0.98; *p* = 0.027) with SGLT2i use. In addition, this pooled meta-analysis suggested that the use of SGLT2is among diabetic patients reduces the risk of cardiovascular death or HHF regardless of the use of angiotensin receptor–neprilysin inhibitors (ARNIs). Patients treated with SGLT2is and ARNIs should be given urgent attention because of the risk of hypotension and renal function impairment.

A subanalysis study of the DECLARE-TIMI 58 trial identified cardiovascular benefits among subjects with reduced ejection fraction, those with HF without known reduced ejection fraction, and those without a history of HF. The study showed a greater risk reduction in cardiovascular death and all-cause mortality among patients of HFrEF treated with dapagliflozin in comparison with those in the other two groups. Patients of HF with or without reduced ejection fraction consistently have outcome benefits in terms of HHF [58]. Another prespecified study of DECLARE-TIMI 58 concluded that dapagliflozin treatment decreases the risk of cardiovascular death and HHF and that the benefits tend to be evident in patients with higher levels of high sensitivity to TnT and N-terminal pro-brain natriuretic peptide [59]. The DEFINE-HE trial, an RCT aimed toward evaluating the dapagliflozin effect on patients with chronic and stable reduced ejection fraction (LVEF ≤ 40%), demonstrated that SGLT2is causes clinically meaningful improvements in quality of life or a reduction in natriuretic peptides [60].

## 7. Sodium–Glucose Cotransporter 2 Inhibitors in Heart Failure with Preserved Ejection Fraction

Two post hoc analysis studies aimed to assess the efficacy of SGLT2is in diabetic patients with HFpEF. The subgroup analysis of the CANVAS program [61] failed to show the positive benefits of canagliflozin for HHF or mortality, although the trend of outcome improvement for the canagliflozin-treated group was observed. The other subgroup analysis of the DECLARE-TIMI 58 study [58] confirmed the reduction in HFH risk in subjects with HFpEF. The results from existing studies should be interpreted with caution because of the small subpopulations of the study groups. Recently, dapagliflozin or canagliflozin improved the Kansas City Cardiomyopathy Questionnaire Clinical Summary Score (a measure of heart failure-related health status) after 12 weeks treatment in patients with HFpEF [62,63]. Empaglifozin also reduced the risk of HHF and cardiovascular mortality in patients with HFpEF [64]. In future, the DELIVER trial will provide further evidence for dapaglifozin in patients with HFpEF [65].

## 8. Updated Guidelines for Sodium–Glucose Cotransporter 2 Inhibitors

Given the substantial evidence from large-scale RCTs and real-world results, SGLT2is is deemed a valuable therapy for the treatment of and reduction in HHF risk among patients with diabetes with underlying HF history. SGLT2is also decreases the risk of HHF or cardiovascular death among patients with HFrEF and those with or without diabetes. In 2020, the American Diabetes Association suggested that SGLT2is are preferred over glucagon-like peptide 1 receptor agonists as a second-line option added to metformin for patients with diabetes with HF history [66]. The 2021 Update to the 2017 American College of Cardiology Expert Consensus [12] suggested SGLT2is as an addition to the therapy regimen for patients with chronic HFrEF who are already receiving guideline-directed medical therapy, such as beta-blockers, ARNI/angiotensin-converting enzyme inhibitors/angiotensin II receptor blockers, and aldosterone antagonists. Moreover, the 2022 AHA/ACC/HFSA guideline also stated the use of SGLT2i is recommended for the management of hyperglycemia and to reduce heart failure-related morbidity and mortality in patients with heart failure and type 2 diabetes [67].

## 9. Conclusions

Given the bidirectional relationship between diabetes and HF and their prognostic implications, emergency service providers should be aware of diabetes-associated HF or the possibility of diabetic cardiomyopathy. At present, the pathophysiology and definition of diabetic cardiomyopathy remain unclear. As recent clinical trial data support the use of SGLT2is in a broad spectrum of HF entities and a comprehensive understanding of the role of SGLT2is in diabetic patients with HF based on cardiovascular outcome trials should help emergency and intensive care service providers identify the best choices for the optimal care of these patients in the future.

## Figures and Tables

**Figure 1 jpm-12-01698-f001:**
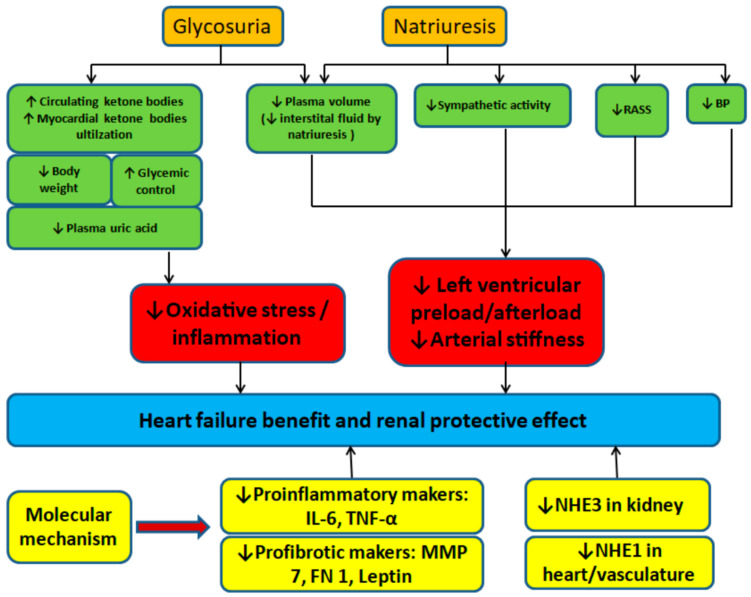
Mechanisms of Heart Failure and Renoprotective Benefits of SGLT2 inhibitor. SGLT2i, Sodium-glucose Cotransporter 2 Inhibitor; RASS, renin-angiotensin-aldosterone system; BP, blood pressure; IL-6, Interleukin 6; TNF, Tumor Necrosis Factor; MMP, Matrix metalloproteinases; FN1, fibronectin 1; NHE, sodium-hydrogen exchanger.

**Table 1 jpm-12-01698-t001:** Summary of randomized controlled trials of SGLT2 inhibitors.

	EMPA-REG Outcome (2015)	CANVAS Program (2017)	DECLARE-TIMI 58 (2019)	VERTIS-CV (2020)
tudy population	T2D and established CVD	T2D and high CV risk	T2D and high CV risk	T2D and established ASCVD
Study numbers (n)	7020	10,142	17,160	8246
Drug	Empagliflozin 10 mg or 25 mg	Canagliflozin 100 mg or 300 mg	Dapagliflozin 10 mg	Ertugliflozin 5 mg or 15 mg
Age (years)	63.1 ± 8.6	63.3 ± 8.3	63.9 ± 6.8	64.4 ± 8.1
Male (%)	71.2	64.2	63.1	70.3
Median follow up (years)	3.1	2.4	4.2	3.5
ASCVD (%)	99.4	72.2	40.5	100
CAD (%)	75.6	56.4	32.9	75.4
Heart failure (%)	9.9	14.4	9.9	23.4
HbA1c (%)	8.07 ± 0.85	8.2 ± 0.9	8.3 ± 1.2	8.2 ± 1.0
Baseline eGFR	74	77	85	76.1 ± 20.9
Body mass index	30.6 ± 5.3	32.0 ± 5.9	32.1 ± 6.0	31.9 ± 5.4
Primary outcome, HR (95% CI)	MACE 0.86 (0.74–0.99) (*p* = 0.04 for superiority)	MACE 0.86 (0.75–0.97) (*p* = 0.02 for superiority)	MACE 0.93 (0.84–1.03) (*p* = 0.17 for superiority)	MACE 0.97 (0.85–1.11) (*p* < 0.001 for noninferiority)
HHF, HR (95% CI)	0.65 (0.50–0.85)	0.67 (0.52–0.87)	0.73 (0.61−0.88)	0.70 (0.54-0.90)
CV death, HR (95% CI)	0.62 (0.49–0.77)	0.87 (0.72–1.06)	0.98 (0.82−1.17)	0.92 (0.77–1.11)
All cause death HR (95% CI)	0.68 (0.57–0.82)	0.87 (0.74–1.01)	0.93 (0.82–1.04)	0.93 (0.80–1.08)
Nonfatal MI, HR (95% CI)	0.87 (0.70–1.09)	0.85 (0.69–1.05)	Fatal/nonfatal MI 0.89 (0.77−1.01)	Fatal/nonfatal MI 1.04 (0.76–1.32)
Nonfatal stroke, HR (95% CI)	1.24 (0.92–1.67)	0.90 (0.71–1.15)	Fatal or nonfatal 1.01 (0.84−1.21)	1.00 (0.76-1.32)
Genital infection Intervention/Placebo	Male: 5.0%/1.5% Female: 10.0%/2.6% (Both *p* < 0.001)	Event rate (per 1000 patient-yr): Female: 68.8/17.5 (*p* < 0.001)	HR (95% CI): 8.36 (4.19–16.68) (*p* < 0.001)	Risk difference (95% CI) Ertugliflozin 5 mg Female:3.6 (1.8–5.7) (*p* < 0.001) Ertugliflozin 15 mg Female:5.4 (3.4–7.7) (*p* < 0.001)
Diabetic ketoacidosis Intervention/Placebo	0.1%/<0.1% No significant differences	Event rate (per 1000 patient-yr): 0.6/0.3 (*p* = 0.14)	HR (95% CI): 2.18 (1.10–4.30) (*p* = 0.02)	Ertugliflozin 5 mg/15 mg/placebo 0.3%/0.4%/0.1%
Bone fracture	3.8 vs. 3.9 No significant differences	Event rate (per 1000 patient-yr): 15.4/11.9(*p* = 0.02)	HR (95% CI): 1.04 (0.91–1.18) (*p* = 0.59)	Ertugliflozin 5 mg/15 mg/placebo 3.6%/3.7%/3.6%

T2D, type 2 diabetes; CVD, cardiovascular disease; ASCVD, atherosclerotic cardiovascular disease; CAD, coronary artery disease; eGFR, estimated glomerular filtration rate (ml/min/1.73 m^2^); Body mass index (kg/m^2^); HR, hazard ratio; CI, confidence interval; MACE, major adverse cardiac event; HHF, hospitalization for heart failure; MI, myocardial infarction.

## Data Availability

Not applicable.

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
