# Peer review of "Diabetes Mellitus and Heart Failure"

_jpm, 2022, doi:10.3390/jpm12101698_

Round 1

Reviewer 1 Report

This review tries to highlight the interactions between diabetes, heart failure and the new drugs with already a class 1 indication for heart failure in the ESC guidelines. Although the authors have performed an extensive literature research this article does not bring any new information and the most important issue of diabetic cardiomyopathy is stil a hot topic among researchers.

There is some minor Ennglish editing to be done. For example line 32 "the prevalence of diabetes is continues to increase" 

I would restructure the paragraphs 4 and 5 as subchapters of chapter 3, because diabetic cardiomyopathy is still under debate if it is a distinct entity or something that overlaps in wide spectrum heart failure syndromes.

Author Response

There is some minor English editing to be done. For example line 32 "the prevalence of diabetes is continues to increase"

ANS: Thank you for your comments! We have been revised it by a native English profession.

I would restructure the paragraphs 4 and 5 as subchapters of chapter 3, because diabetic cardiomyopathy is still under debate if it is a distinct entity or something that overlaps in wide spectrum heart failure syndromes.

ANS: Thank you for your comments! We have been restructured the paragraphs 4 and 5 as subchapters of chapter 3 (see page 6-11).

Reviewer 2 Report

The article covers an interesting topic and summarizes important clinical-trial data on the cardioprotective effect of SGLT-2 inhibitors in patients with T2D and heart failure. I have some additional suggestions to the authors:

   1.                In the SGLT-2 inhibitor section, the CREDENCE trial and DAPA-CKD trial also showed an impressive reduction in the risk of HF hospitalization. Please add these trials to your discussion.

   2.                The review could be improved with the addition of data on the efficacy and safety of MRAs. The steroidal MRAs spironolactone and eplerenone are guideline-directed therapies in patients with HFrEF. The novel non-steroidal MRA finerenone was shown to effectively reduce the risk of HF hospitalization in patients with T2D and a broad spectrum of CKD. The results of the FIDELIO-DKD and FIGARO-DKD trials need to be discussed in this article. The anti-inflammatory and anti-fibrotic actions of MR antagonism represent an important pathway to protect the heart in patients with T2D.

Author Response

  1. In the SGLT-2 inhibitor section, the CREDENCE trial and DAPA-CKD trial also showed an impressive reduction in the risk of HF hospitalization. Please add these trials to your discussion.

ANS: Thank you for your comments! We have added these trials to the discussion (see page 13-14).

  1. The review could be improved with the addition of data on the efficacy and safety of MRAs. The steroidal MRAs spironolactone and eplerenone are guideline-directed therapies in patients with HFrEF. The novel non-steroidal MRA finerenone was shown to effectively reduce the risk of HF hospitalization in patients with T2D and a broad spectrum of CKD. The results of the FIDELIO-DKD and FIGARO-DKD trials need to be discussed in this article. The anti-inflammatory and anti-fibrotic actions of MR antagonism represent an important pathway to protect the heart in patients with T2D.

ANS: Thank you for your comments! We have been added these trials to a new paragraph (see page 11).

Reviewer 3 Report

This is an interesting article which reviews relationshis between heart failure and diabetes mellitus including pathophysiological processess underlying this association, pathophysiology of diabetic cardiomyopathy, prognostic implications of this reletionship as well as treatment with different classess of glucose lowering drugs, especially the role of sodium glucose cotransporter 2 inhibitors (SGLT2Is) in diabetes and HF with preserved and reduced ejestion fraction and this population.

Author Response

Thank you for your comments!

Reviewer 4 Report

Dear Authors, dear Editorial Office,

Thank you for the opportunity to review the paper „Diabetes Mellitus and Heart Failure” by Wun-Zhih et al. 

The paper nicely discusses the role of SGLT2 inhibitors in heart failure both systolic and diastolic. It lists ample number of RCT to emphasizes the role of these agents in T2DM and patients without diabetes. Pathomechanism is also discussed to an acceptable level.

When looking up the subject in PubMed, this is what we find:

https://pubmed.ncbi.nlm.nih.gov/?term=diabetes+heart+failure+sglt2&sort=date

117 papers, newest from 2022, forms are metanalysis, reviews, focused article, etc., partly listed in the references.

Based on the above, two things need to be mentioned:

1.     The paper in its form is of good quality- based on that, publishing it is of good choice.

2.     On the other hand, partial similar papers are already in the scientific field

Since it is a review, very hard to criticize the subject, since all the papers mentioned are already published, meaning research-wise of good quality.

Based on the above, publishing this paper is not against my will, but other reviewer’s opinion should be weighted high.

Kind regards,

Rudolf

Author Response

Thank you for your comments!

Reviewer 5 Report

Chapter 4 does not clear explain the links between insulin resistance and renin angiotensin up-regulation

Chapter 5 There is no clear explanations of the origin and significance of AGEs. 

Author Response

Chapter 4 does not clear explain the links between insulin resistance and renin angiotensin up-regulation

ANS: Thank you for your comments! We have been revised it (see line 162-168, page 8-9).

Chapter 5 There is no clear explanations of the origin and significance of AGEs.

ANS: Thank you for your comments! We have been revised it (see line 182-193, page 9-10).

Round 2

Reviewer 1 Report

Compared to the initial version, the review has been revised and improved and is now in a publishable form.